# Understanding and Mitigating the Limitations of Prioritized Experience Replay

**Yangchen Pan**[1,3]        **Jincheng Mei**[*1]        **Amir-massoud Farahmand**[2,4]        **Martha White**[1,4]

**Hengshuai Yao**[1]        **Mohsen Rohani**[3]        **Jun Luo**[3]

[1]University of Alberta
[2]University of Toronto & Vector Institute
[3]Huawei Noah's Ark Lab
[4]CIFAR AI Chair

## Abstract

Prioritized Experience Replay (ER) has been empirically shown to improve sample efficiency across many domains and attracted great attention; however, there is little theoretical understanding of why such prioritized sampling helps and its limitations. In this work, we take a deep look at the prioritized ER. In a supervised learning setting, we show the equivalence between the error-based prioritized sampling method for minimizing mean squared error and the uniform sampling for cubic power loss. We then provide theoretical insight into why error-based prioritized sampling improves convergence rate upon uniform sampling when minimizing mean squared error during early learning. Based on the insight, we further point out two limitations of the prioritized ER method: 1) outdated priorities and 2) insufficient coverage of the sample space. To mitigate the limitations, we propose our model-based stochastic gradient Langevin dynamics sampling method. We show that our method does provide states distributed close to an ideal prioritized sampling distribution estimated by the brute-force method, which does not suffer from the two limitations. We conduct experiments on both discrete and continuous control problems to show our approach's efficacy and examine the practical implication of our method in an autonomous driving application.

## 1  INTRODUCTION

Experience Replay (ER) [Lin, 1992] has been a popular method for training large-scale modern Reinforcement Learning (RL) systems [Degris et al., 2012, Adam and Busoniu, 2012, Mnih et al., 2015a, Hessel et al., 2018, François-Lavet et al., 2018]. In ER, visited experiences are stored in a buffer, and at each time step, a mini-batch of experiences is *uniformly* sampled to update the training parameters in the value or policy function. Such a method is empirically shown to effectively stabilize the training and improve the sample efficiency of deep RL algorithms. Several follow-up works propose to improve upon it by designing non-uniform sampling distributions or re-weighting mechanisms of experiences [Schaul et al., 2016, Andrychowicz et al., 2017, Oh et al., 2018, de Bruin et al., 2018, Horgan et al., 2018, Zha et al., 2019, Novati and Koumoutsakos, 2019, Kumar et al., 2020, Sun et al., 2020, Liu et al., 2021, Lee et al., 2021, Sinha et al., 2022]. The most relevant one to our work is prioritized ER [Schaul et al., 2016], which attempts to improve the vanilla ER method by sampling those visited experiences proportional to their absolute Temporal Difference (TD) errors. Empirically, it can significantly improve sample efficiency upon vanilla ER on many domains.

ER methods have a close connection to Model-Based RL (MBRL) methods [Kaelbling et al., 1996, Bertsekas, 2009, Sutton and Barto, 2018]. ER can be thought of as an instance of a classical model-based RL architecture—Dyna [Sutton, 1991], using a non-parametric model given by the buffer [van Seijen and Sutton, 2015, van Hasselt et al., 2019]. A Dyna agent uses real experience to update its policy as well as its reward and dynamics model. In-between taking actions, the agent can get hypothetical experiences from the model and use them just like the real experiences to further improve the policy. How to generate those hypothetical experiences is largely dependent on **search-control**—the mechanism of generating states or state-action pairs from which to query the model to get the next states and rewards. Existing works show that smart search-control strategies can further improve sample efficiency of a Dyna agent [Sutton et al., 2008, Gu et al., 2016, Goyal et al., 2019, Holland et al., 2018, Pan et al., 2018, Corneil et al., 2018, Janner et al., 2019, Chelu et al., 2020]. Particularly, prioritized

---

*Work done while at the University of Alberta. Equal contribution with Yangchen Pan. Correspondence to: Yangchen Pan <pan6@ualberta.ca> and Jincheng Mei <jmei2@ualberta.ca>.

*Accepted for the 38th Conference on Uncertainty in Artificial Intelligence* (UAI 2022).

sweeping [Moore and Atkeson, 1993] is among the earliest work that improves upon vanilla Dyna. The idea behind prioritized sweeping is quite intuitive: we should give high priority to states whose absolute TD errors are large because they are likely to cause the most change in value estimates. Hence, the prioritized ER by Schaul et al. [2016], which applies TD error-based prioritized sampling to ER, is a natural idea in a model-free RL setting. However, there is little rigorous understanding towards prioritized ER method about why it can help and its limitations.

This work provides a theoretical insight into the prioritized ER's advantage and points out its two drawbacks: outdated priorities and insufficient sample space coverage, which may significantly weaken its efficacy. To mitigate the two issues, we propose to use the Stochastic Gradient Langevin Dynamics (SGLD) sampling method to acquire states. Our method relies on applying an environment model to 1) simulate priorities of states and 2) acquire hypothetical experiences. Then these experiences are used for further improving the policy. We demonstrate that, comparing with the conventional prioritized ER method, the hypothetical experiences generated by our method are distributed closer to the *ideal* TD error-based sampling distribution, which does not suffer from the two drawbacks. Finally, we demonstrate the utility of our approach on various benchmark discrete and continuous control domains and an autonomous driving application.

## 2 BACKGROUND

In this section, we firstly review basic concepts in RL. Then we briefly introduce the prioritized ER method, which will be examined in-depth in the next section. We conclude this section by discussing a classic MBRL architecture called Dyna [Sutton, 1991] and its recent variants, which are most relevant to our work.

**Basic notations.** We consider a discounted Markov Decision Process (MDP) framework [Szepesvári, 2010]. An MDP can be denoted as a tuple $(\mathcal{S}, \mathcal{A}, \mathbb{P}, R, \gamma)$ including state space $\mathcal{S}$, action space $\mathcal{A}$, probability transition kernel $\mathbb{P}$, reward function $R$, and discount rate $\gamma \in [0, 1]$. At each environment time step $t$, an RL agent observes a state $s_t \in \mathcal{S}$, takes an action $a_t \in \mathcal{A}$, and moves to the next state $s_{t+1} \sim \mathbb{P}(\cdot|s_t, a_t)$, and receives a scalar reward signal $r_{t+1} = R(s_t, a_t, s_{t+1})$. A policy is a mapping $\pi : \mathcal{S} \times \mathcal{A} \to [0, 1]$ that determines the probability of choosing an action at a given state.

A popular algorithm to find an optimal policy is Q-learning [Watkins and Dayan, 1992]. With function approximation, parameterized action-values $Q_\theta$ are updated using $\theta = \theta + \alpha \delta_t \nabla_\theta Q_\theta(s_t, a_t)$ for stepsize $\alpha > 0$ with TD-error $\delta_t \overset{\text{def}}{=} r_{t+1} + \gamma \max_{a' \in \mathcal{A}} Q_\theta(s_{t+1}, a') - Q_\theta(s_t, a_t)$. The policy is defined by acting greedily w.r.t. these action-values.

**ER methods.** ER is critical when using neural networks to estimate $Q_\theta$, as used in DQN [Mnih et al., 2015b], both to stabilize and speed up learning. The vanilla ER method uniformly samples a mini-batch of experiences from those visited ones in the form of $(s_t, a_t, s_{t+1}, r_{t+1})$ to update neural network parameters. Prioritized ER [Schaul et al., 2016] improves upon it by sampling prioritized experiences, where the probability of sampling a certain experience is proportional to its TD error magnitude, i.e., $p(s_t, a_t, s_{t+1}, r_{t+1}) \propto |\delta_t|$. However, the underlying theoretical mechanism behind this method is still not well understood.

**MBRL and Dyna.** With a model, an agent has more flexibility to sample hypothetical experiences. We consider a one-step model which takes a state-action pair as input and provides a distribution over the next state and reward. We build on the Dyna formalism [Sutton, 1991] for MBRL, and more specifically, the recently proposed (Hill Climbing) HC-Dyna [Pan et al., 2019] as shown in Algorithm 1. HC-Dyna provides some smart approach to *Search-Control* (SC).

HC-Dyna proposes to employ stochastic gradient Langevin dynamics (SGLD) for sampling states, which relies on hill climbing on some criterion function $h(\cdot)$. The term "Hill Climbing (HC)" is used for generality as the SGLD sampling process can be thought of as doing some modified gradient ascent [Pan et al., 2019, 2020].

The algorithmic framework maintains two buffers: the conventional ER buffer storing experiences (i.e., an experience/transition has the form of $(s_t, a_t, s_{t+1}, r_{t+1})$) and a *search-control queue* storing the states acquired by search-control mechanisms (i.e., SLGD sampling). At each time step $t$, a real experience $(s_t, a_t, s_{t+1}, r_{t+1})$ is collected and stored into the ER buffer. Then the HC search-control process starts to collect states and store them into the search-control queue. A hypothetical experience is obtained by first selecting a state $s$ from the search-control queue, then selecting an action $a$ according to the current policy, and then querying the model to get the next state $s'$ and reward $r$ to form an experience $(s, a, s', r)$. These hypothetical transitions are combined with real experiences into a single mini-batch to update the training parameters. The $n$ updates, performed before taking the next action, are called *planning updates* [Sutton and Barto, 2018], as they improve the value/policy by using a model. The choice of pairing states with on-policy actions to form hypothetical experiences has been reported to be beneficial [Gu et al., 2016, Pan et al., 2018, Janner et al., 2019].

Two instances have been proposed for $h(\cdot)$: the value function $v(s)$ [Pan et al., 2019] and the sum of gradient and Hessian magnitude $||\nabla_s v(s)|| + ||H_v(s)||$ [Pan et al., 2020]. The former is used as a measure of the utility of a state: doing HC on the learned value function should find high-value states without being constrained by the physical environment dynamics. The latter is considered as a measure of

**Algorithm 1** HC-Dyna: Generic framework

**Input:** Hill Climbing (HC) criterion function $h : \mathcal{S} \mapsto \mathbb{R}$; batch-size $b$; initialize empty search-control queue $B_{sc}$; empty ER buffer $B_{er}$; initialize policy and model $\mathcal{P}$; HC stepsize $\alpha_h$; mini-batch size $b$; environment $\mathcal{P}$; mixing rate $\rho$ decides the proportion of hypothetical experiences in a mini-batch.

**for** $t = 1, 2, \ldots$ **do**
    Add $(s_t, a_t, s_{t+1}, r_{t+1})$ to $B_{er}$
    **while** within some budget time steps **do**
        // SGLD sampling for states
        $s \leftarrow s + \alpha_h \nabla_s h(s)$ + Gaussian noise // Search-control, see Section 4 for details about SGLD sampling
        Add $s$ into $B_{sc}$
    // $n$ planning updates/steps
    **for** $n$ times **do**
        $B \leftarrow \emptyset$ // initialize an empty mini-batch $B$
        **for** $b\rho$ times **do**
            Sample $s \sim B_{sc}$, on-policy action $a$
            Sample $s', r \sim \mathcal{P}(s, a)$
            Add $(s, a, s', r)$ into $B$
        Sample $b(1 - \rho)$ experiences from $B_{er}$, add to $B$
        // NOTE: if $\rho = 0$, then we only uniformly sample $b$ experiences from $B_{er}$ and use them as $B$, and the algorithm reduces to ER
        Update policy/value on mixed mini-batch $B$

the value approximation difficulty, then doing HC provides additional states whose values are difficult to learn. The two suffer from several issues as we discuss in the Appendix A.1. This paper will introduce a HC search-control method motivated by overcoming the limitations of the prioritized ER.

# 3 A DEEPER LOOK AT ERROR-BASED PRIORITIZED SAMPLING

In this section, we provide theoretical motivation for error-based prioritized sampling by showing its equivalence to optimizing a cubic power objective with uniform sampling in a supervised learning setting. We prove that optimizing the cubic objective provides a faster convergence rate during early learning. Based on the insight, we discuss two limitations of the prioritized ER: 1) outdated priorities and 2) insufficient coverage of the sample space. We then empirically study the limitations.

## 3.1 THEORETICAL INSIGHT INTO ERROR-BASED PRIORITIZED SAMPLING

In the $l_2$ regression, we minimize the mean squared error $\min_\theta \frac{1}{2n} \sum_{i=1}^n (f_\theta(x_i) - y_i)^2$, for training set $\mathcal{T} = \{(x_i, y_i)\}_{i=1}^n$ and function approximator $f_\theta$, such as a neu-

ral network. In error-based prioritized sampling, we define the priority of a sample $(x, y) \in \mathcal{T}$ as $|f_\theta(x) - y|$; the probability of drawing a sample $(x, y) \in \mathcal{T}$ is typically $q(x, y; \theta) \propto |f_\theta(x) - y|$. We employ the following form to compute the probability of a point $(x, y) \in \mathcal{T}$:

$$q(x, y; \theta) \stackrel{\text{def}}{=} \frac{|f_\theta(x) - y|}{\sum_{i=1}^n |f_\theta(x_i) - y_i|}. \tag{1}$$

We can show an equivalence between the gradients of the squared objective with this prioritization and the cubic power objective $\frac{1}{3n} \sum_{i=1}^n |f_\theta(x_i) - y_i|^3$ in Theorem 1 below. See Appendix A.3 for the proof.

**Theorem 1.** *For a constant $c$ determined by $\theta, \mathcal{T}$, we have*

$$c\mathbb{E}_{(x,y) \sim q(x,y;\theta)}[\nabla_\theta (1/2)(f_\theta(x) - y)^2]$$
$$= \mathbb{E}_{(x,y) \sim uniform(\mathcal{T})}[\nabla_\theta (1/3)|f_\theta(x) - y|^3].$$

We empirically verify this equivalence in the Appendix A.7. This simple theorem provides an intuitive reason for why prioritized sampling can help improve sample efficiency: the gradient direction of the cubic function is sharper than that of the square function when the error is relatively large (Figure 8). We refer readers to the work by Fujimoto et al. [2020] regarding more discussions about the equivalence between prioritized sampling and of uniform sampling. Theorem 2 below further proves that optimizing the cubic power objective by gradient descent has faster convergence rate than the squared objective, and this provides a solid motivation for using error-based prioritized sampling. See Appendix A.4 for a detailed version of the theorem below, and its proof and empirical simulations.

**Theorem 2** (Fast early learning, concise version)**.** *Let $n$ be a positive integer (i.e., the number of training samples). Let $x_t, \tilde{x}_t \in \mathbb{R}^n$ be the target estimates of all samples at time $t, t \geq 0$, and $x(i)(i \in [n], [n] \stackrel{\text{def}}{=} \{1, 2, ..., n\})$ be the $i$th element in the vector. We define the objectives:*

$$\ell_2(x, y) \stackrel{\text{def}}{=} \frac{1}{2} \sum_{i=1}^n (x(i) - y(i))^2,$$

$$\ell_3(x, y) \stackrel{\text{def}}{=} \frac{1}{3} \sum_{i=1}^n |x(i) - y(i)|^3.$$

*Let $\{x_t\}_{t \geq 0}$ and $\{\tilde{x}_t\}_{t \geq 0}$ be generated by using $\ell_2, \ell_3$ objectives respectively. Then define the total absolute prediction errors respectively:*

$$\delta_t \stackrel{\text{def}}{=} \sum_{i=1}^n \delta_t(i) = \sum_{i=1}^n |x_t(i) - y(i)|,$$

$$\tilde{\delta}_t \stackrel{\text{def}}{=} \sum_{i=1}^n \tilde{\delta}_t(i) = \sum_{i=1}^n |\tilde{x}_t(i) - y(i)|,$$

where $y(i) \in \mathbb{R}$ is the training target for the $i$th training sample. That is, $\forall i \in [n]$,

$$\frac{dx_t(i)}{dt} = -\eta \cdot \frac{d\ell_2(x_t, y)}{dx_t(i)}, \quad \frac{d\tilde{x}_t(i)}{dt} = -\eta' \cdot \frac{d\ell_3(\tilde{x}_t, y)}{d\tilde{x}_t(i)}.$$

Given any $0 < \epsilon \leq \delta_0 = \sum_{i=1}^n \delta_0(i)$, define the following hitting time,

$$t_\epsilon \stackrel{\text{def}}{=} \min_t\{t \geq 0 : \delta_t \leq \epsilon\}, \quad \tilde{t}_\epsilon \stackrel{\text{def}}{=} \min_t\{t \geq 0 : \tilde{\delta}_t \leq \epsilon\}.$$

Assume the same initialization $x_0 = \tilde{x}_0$. **We have the following conclusion.**
If there exists $\delta_0 \in \mathbb{R}$ and $0 < \epsilon \leq \delta_0$ such that

$$\frac{1}{n} \cdot \sum_{i=1}^n \frac{1}{\delta_0(i)} \leq \frac{\eta}{\eta'} \cdot \frac{\log(\delta_0/\epsilon)}{\frac{\delta_0}{\epsilon} - 1}, \quad (2)$$

then we have $t_\epsilon \geq \tilde{t}_\epsilon$, which means gradient descent using the cubic loss function will achieve the total absolute error threshold $\epsilon$ faster than using the squared objective function.

This theorem illustrates that when the total loss of all training examples is greater than some threshold, cubic power learns faster. For example, let the number of samples $n = 1000$, and each sample has initial loss $\delta_0(i) = 2$. Then $\delta_0 = 2000$. Setting $\epsilon = 570$ (i.e., $\epsilon(i) \approx 0.57$) satisfies the inequality (2). This implies that using the cubic objective is faster in reducing the total loss from 2000 to 570. Though it is not our focus here to investigate the practical utility of the high power objectives, we include some empirical results and discuss the practical utilities of such objectives in Appendix A.6.

Note that, although the original prioritized ER raises the importance ratio to a certain power, which is annealing from 1 at the beginning to 0 [Schaul et al., 2016]; our theorem still explains the improvement of sample efficiency during the early learning stage. It is because, the power is close to one and hence it is equivalent to using a higher power loss. This point has also been confirmed by a concurrent work [Fujimoto et al., 2020, Sec 5.1, Theorem 3].

## 3.2 LIMITATIONS OF THE PRIORITIZED ER

Inspired by the above theorems, we now discuss two drawbacks of prioritized sampling: **outdated priorities** and **insufficient sample space coverage**. Then we empirically examine their importance and effects in the next section.

The above two theorems show that the advantage of prioritized sampling comes from the faster convergence rate of cubic power objective during early learning. By Theorem 1, such advantage requires to update the priorities of *all training samples* by using the *updated training parameters $\theta$* at each time step. In RL, however, at the each time step $t$, the original prioritized ER method only updates the priorities of those experiences from the sampled mini-batch, leaving the priorities of the rest of experiences unchanged [Schaul et al., 2016]. We call this limitation **outdated priorities**. It is typically infeasible to update the priorities of all experiences at each time step.

In fact, in RL, "*all training samples*" in RL are restricted to those visited experiences in the ER buffer, which may only contain a small subset of the whole state space, making the estimate of the prioritized sampling distribution inaccurate. There can be many reasons for the small coverage: the exploration is difficult, the state space is huge, or the memory resource of the buffer is quite limited, etc. We call this issue **insufficient sample space coverage**, which is also noted by Fedus et al. [2020].

Note that *insufficient sample space coverage* should not be considered equivalent to off-policy distribution issue. The latter refers to some old experiences in the ER buffer may be unlikely to appear under the current policy [Novati and Koumoutsakos, 2019, Zha et al., 2019, Sun et al., 2020, Oh et al., 2021]. In contrast, the issue of insufficient sample space coverage can raise naturally. For example, the state space is large and an agent is only able to visit a small subset of the state space during early learning stage. We visualize the state space coverage issue on a RL domain in Section 4.

## 3.3 NEGATIVE EFFECTS OF THE LIMITATIONS

In this section, we empirically show that the outdated priorities and insufficient sample space coverage significantly blur the advantage of the prioritized sampling method.

**Experiment setup**. We conduct experiments on a supervised learning task. We generate a training set $\mathcal{T}$ by uniformly sampling $x \in [-2, 2]$ and adding zero-mean Gaussian noise with standard deviation $\sigma = 0.5$ to the target $f_{\sin}(x)$ values. Define $f_{\sin}(x) \stackrel{\text{def}}{=} \sin(8\pi x)$ if $x \in [-2, 0)$ and $f_{\sin}(x) = \sin(\pi x)$ if $x \in [0, 2]$. The testing set contains 1k samples where the targets are not noise-contaminated. Previous work [Pan et al., 2020] shows that the high frequency region $[-2, 0]$ usually takes long time to learn. Hence we expect error-based prioritized sampling to make a clear difference in terms of sample efficiency on this dataset. We use $32 \times 32$ tanh layers neural network for all algorithms. We refer to Appendix A.8 for missing details and A.7 for additional experiments.

**Naming of algorithms**. **L2**: the $l_2$ regression with uniformly sampling from $\mathcal{T}$. **Full-PrioritizedL2**: the $l_2$ regression with prioritized sampling according to the distribution defined in (1), the priorities of *all samples* in the training set are updated after each mini-batch update. **PrioritizedL2**: the only difference with **Full-PrioritizedL2** is that *only* the priorities of those training examples sampled in the mini-batch are updated at each iteration, the rest of the training samples use the original priorities. This resembles the ap-

proach taken by the prioritized ER in RL [Schaul et al., 2016]. We show the learning curves in Figure 1.

**Outdated priorities.** Figure 1 (a) shows that PrioritizedL2 without updating all priorities can be significantly worse than Full-PrioritizedL2. Correspondingly, we further verify this phenomenon on the classical Mountain Car domain [Brockman et al., 2016]. Figure 1(c) shows the evaluation learning curves of different DQN variants in an RL setting. We use a small $16 \times 16$ ReLu NN as the $Q$-function, which should highlight the issue of priority updating: every mini-batch update potentially perturbs the values of many other states. Hence many experiences in the ER buffer have the wrong priorities. Full-PrioritizedER does perform significantly better.

**Sample space coverage.** To check the effect of insufficient sample space coverage, we examine how the relative performances of L2 and Full-PrioritizedL2 change when we train them on a smaller training dataset with only $400$ examples as shown in Figure 1(b). The small training set has a small coverage of the sample space. Unsurprisingly, using a small training set makes all algorithms perform worse; however, *it significantly narrows the gap between Full-PrioritizedL2 and L2*. This indicates that prioritized sampling needs sufficient samples across the sample space to estimate the prioritized sampling distribution reasonably accurate. We further verify the sample space coverage issue in prioritized ER on a RL problem in the next section.

# 4 ADDRESSING THE LIMITATIONS

In this section, we propose a Stochastic Gradient Langevin Dynamics (SGLD) sampling method to mitigate the limitations of the prioritized ER method mentioned in the above section. Then we empirically examine our sampling distribution. We also describe how our sampling method is used for the search-control component in Dyna.

## 4.1 SAMPLING METHOD

**SGLD sampling method.** Let $v^\pi(\cdot; \theta) : \mathcal{S} \mapsto \mathbb{R}$ be a differentiable value function under policy $\pi$ parameterized by $\theta$. For $s \in \mathcal{S}$, define $y(s) \overset{\text{def}}{=} \mathbb{E}_{r,s' \sim \mathcal{P}^\pi(s', r|s)}[r + \gamma v^\pi(s'; \theta)]$, and denote the TD error as $\delta(s, y; \theta_t) \overset{\text{def}}{=} y(s) - v(s; \theta_t)$. Given some initial state $s_0 \in \mathcal{S}$, let the state sequence $\{s_i\}$ be the one generated by updating rule $s_{i+1} \leftarrow s_i + \alpha_h \nabla_s \log |\delta(s_i, y(s_i); \theta_t)| + X_i$, where $\alpha_h$ is a stepsize and $X_i$ is a Gaussian random variable with some constant variance.[1] Then $\{s_i\}$ converges to the distribution $p(s) \propto |\delta(s, y(s))|$ as $i \to \infty$. The proof is a direct consequence of the convergent behavior of Langevin dynam-

---

[1] The stepsize and variance decides the temperature parameter in the Gibbs distribution: $2\alpha_h / \sigma^2$ [Zhang et al., 2017]. The two parameters are usually treated as hyper-parameters in practice.

ics stochastic differential equation (SDE) [Roberts, 1996, Welling and Teh, 2011, Zhang et al., 2017]. We include a brief background knowledge in Appendix A.2.

It should be noted that, this sampling method enables us to acquire states *1) whose absolute TD errors are estimated by using the current parameter $\theta_t$ and 2) that are not restricted to those visited ones*. We empirically verify the two points in Section 4.2.

**Implementation**. In practice, we can compute the state value estimate by $v(s) = \max_a Q(s, a; \theta_t)$ as suggested by Pan et al. [2019]. In the case that a true environment model is not available, we compute an estimate $\hat{y}(s)$ of $y(s)$ by a learned model. Then at each time step $t$, states approximately following the distribution $p(s) \propto |\delta(s, y(s))|$ can be generated by

$$s \leftarrow s + \alpha_h \nabla_s \log |\hat{y}(s) - \max_a Q(s, a; \theta_t)| + X, \quad (3)$$

where $X$ is a Gaussian random variable with zero-mean and some small variance. Observing that $\alpha_h$ is small, we consider $\hat{y}(s)$ as a constant given a state $s$ without backpropagating through it. Though this updating rule introduces bias due to the usage of a learned model, fortunately, the difference between the sampling distribution acquired by the true model and the learned model can be upper bounded as we show in Theorem 3 in Appendix A.5.

**Algorithmic details.** We present our algorithm called **Dyna-TD** in the Algorithm 3 in Appendix A.8. Our algorithm follows the general steps in Algorithm 1. Particularly, we choose the function $h(s) \overset{\text{def}}{=} \log |\hat{y}(s) - \max_a Q(s, a; \theta_t)|$ for HC search-control process, i.e., run the updating rule 3 to generate states.

## 4.2 EMPIRICAL VERIFICATION OF TD ERROR-BASED SAMPLING METHOD

We visualize the distribution of the sampled states by our method and those from the buffer of the prioritized ER, verifying that our sampled states have an obviously larger coverage of the state space. We then empirically verify that our sampling distribution is closer to a brute-force calculated prioritized sampling distribution—which does not suffer from the two limitations—than the prioritized ER method. Finally, we discuss concerns regarding computational cost. Please see Appendix A.8 for any missing details.

**Large sample space coverage**. During early learning, we visualize 2k states sampled from 1) DQN's buffer trained by prioritized ER and 2) our algorithm Dyna-TD's Search-Control (SC) queue on the continuous state GridWorld (Figure 2(a)). Figure 2 (b-c) visualize state distributions with different sampling methods via heatmap. Darker color indicates higher density. (b)(c) show that DQN's ER buffer, no matter with or without prioritized sampling, does not

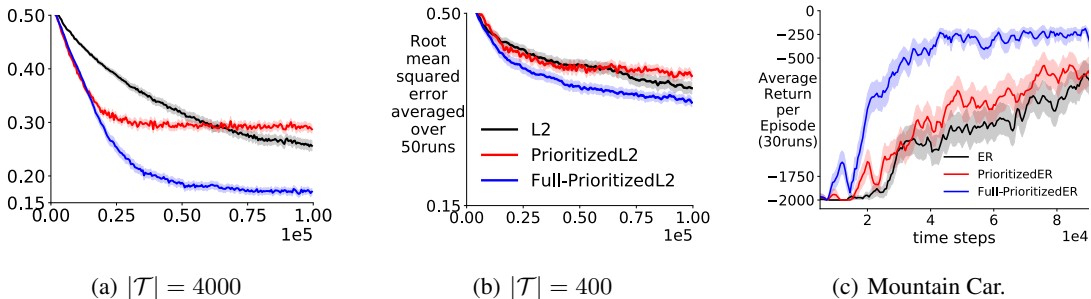

(a) $|\mathcal{T}| = 4000$      (b) $|\mathcal{T}| = 400$      (c) Mountain Car.

Figure 1: Comparing **L2 (black)**, **PrioritizedL2 (red)**, and **Full-PrioritizedL2 (blue)** in terms of testing RMSE v.s. number of mini-batch updates. (a)(b) show the results trained on a large and small training set, respectively. (c) shows the result of a corresponding RL experiment on mountain car domain. We compare episodic return v.s. environment time steps for **ER (black)**, **PrioritizedER (red)**, and **Full-PrioritizedER (blue)**. Results are averaged over 50 random seeds on (a), (b) and 30 on (c). The shade indicates standard error.

cover well the top-left part and the right half part on the GridWorld. In contrast, Figure 2 (d) shows that states from our SC queue are more diversely distributed on the square. These visualizations verify that our sampled states cover better the state space than the prioritized ER does.

**Notations and experiment setting**. We denote our sampling distribution as $p_1(\cdot)$, the one acquired by conventional prioritized ER as $p_2(\cdot)$, and the one computed by thorough priority updating of enumerating all states in the state space as $p^*(\cdot)$ (this one should be unrealistic in practice and we call it the ideal distribution as it does not suffer from the two limitations we discussed). We visualize how well $p_1(\cdot)$ and $p_2(\cdot)$ can approximate $p^*(\cdot)$ on the GridWorld domain, where the state distributions can be conveniently estimated by discretizing the continuous state GridWorld to a $50 \times 50$ one. We compute the distances of $p_1, p_2$ to $p^*$ by two sensible weighting schemes: 1) on-policy weighting: $\sum_{j=1}^{2500} d^\pi(s_j)|p_i(s_j) - p^*(s_j)|, i \in \{1,2\}$, where $d^\pi$ is approximated by uniformly sample 3k states from a recency buffer; 2) uniform weighting: $\frac{1}{2500}\sum_{j=1}^{2500} |p_i(s_j) - p^*(s_j)|, i \in \{1,2\}$.

**Sampling distribution is close to the ideal one**. We plot the distances change when we train our Algorithm 3 and the prioritized ER in Figure 3(a)(b). They show that the HC procedure in our algorithm Dyna-TD-Long produces a state distribution with a significantly closer distance to the desired sampling distribution $p^*$ than PrioritizedER under both weighting schemes. In contrast, the state distribution acquired from PrioritizedER, which suffers from the two limitations, is far away from $p^*$. Note that the suffix "-Long" of Dyna-TD-Long indicates that we run a large number of SGLD steps (i.e., 1k) to reach stationary behavior. This is a sanity check but impractical; hence, we test the version with only a few SGLD steps.

**Sampling distribution with much fewer SGLD steps**. In practice, we probably only want to run a small number of SGLD steps to save time. As a result, we include a practical

version of Dyna-TD, which only runs 30 SGLD steps, with either a true or learned model. Figure 3(a)(b) show that even a few SGLD steps can give better sampling distribution than the conventional PrioritizedER does.

**Computational cost.** Let the mini-batch size be $b$, and the number of HC steps be $k_{HC}$. If we assume one mini-batch update takes $\mathcal{O}(c)$, then the time cost of our sampling is $\mathcal{O}(ck_{HC}/b)$, which is reasonable. On the GridWorld, Figure 3(c) shows that given the same time budget, our algorithm achieves better performance.This makes the additional time spent on search-control worth it.

## 5   EXPERIMENTS

In this section, we firstly introduce baselines and the basic experimental setup. Then we design experiments in the three paragraphs 1) *performances on benchmarks*, 2) *Dyna variants comparison*, and 3) *a demo for continuous control* to answer three following questions correspondingly.

1. By mitigating the limitations of the conventional prioritized ER method, can Dyna-TD outperform the prioritized ER under various planning budgets in different environments?

2. Can Dyna-TD outperform the existing Dyna variants?

3. How effective is Dyna-TD under an online learned model, particularly for more realistic applications where actions are continuous?

**Baselines and basic setup. ER** is DQN with a regular ER buffer without prioritized sampling. **PrioritizedER** is the one by Schaul et al. [2016], which has the drawbacks as discussed in our paper. **Dyna-Value** [Pan et al., 2019] is the Dyna variant which performs HC on the learned value function to acquire states to populate the SC queue. **Dyna-Frequency** [Pan et al., 2020] is the Dyna variant which performs HC on the norm of the gradient of the value function to acquire states to populate the SC queue. For fair

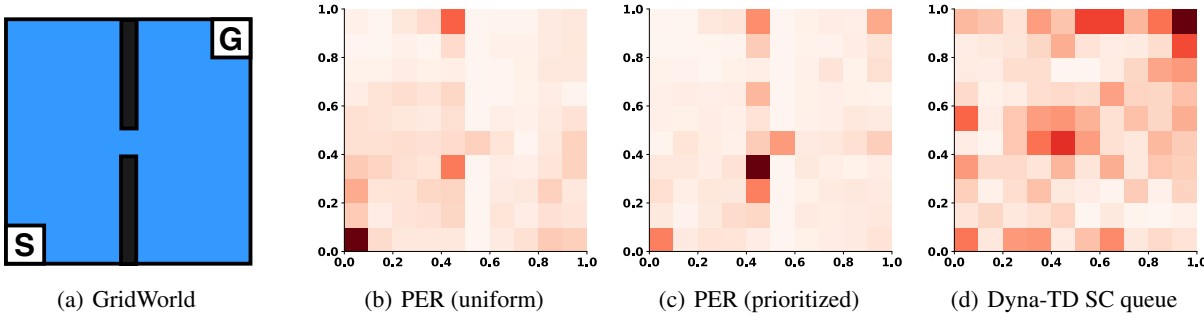

(a) GridWorld  (b) PER (uniform)  (c) PER (prioritized)  (d) Dyna-TD SC queue

Figure 2: (a) shows the GridWorld [Pan et al., 2019]. It has $\mathcal{S} = [0,1]^2, \mathcal{A} = \{up, down, right, left\}$. The agent starts from the left bottom and learn to reach the right top within as few steps as possible. (b) and (c) respectively show the state distributions with uniform and prioritized sampling methods from the ER buffer of prioritized ER. (d) shows the SC queue state distribution of our Dyna-TD. Dark color indicates high density.

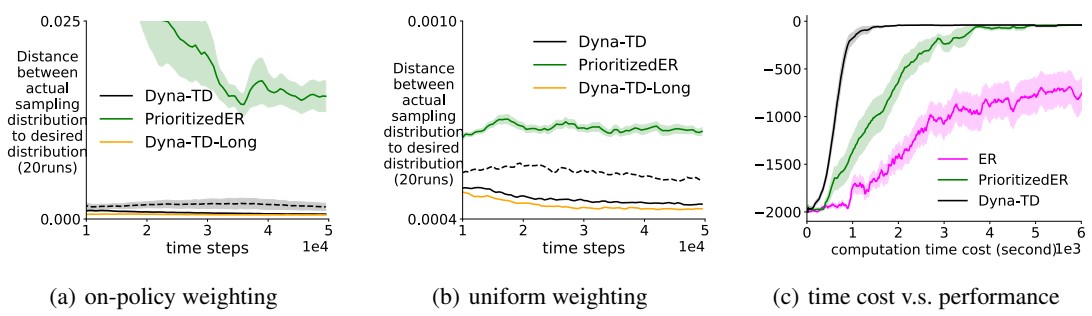

(a) on-policy weighting  (b) uniform weighting  (c) time cost v.s. performance

Figure 3: (a)(b) show the distance change as a function of environment time steps for **Dyna-TD (black)**, **PrioritizedER (forest green)**, and **Dyna-TD-Long (orange)**, with different weighting schemes. The **dashed** line corresponds to our algorithm with an online learned model. The corresponding evaluation learning curve is in the Figure 4(c). (d) shows the policy evaluation performance as a function of running time (in seconds) with **ER(magenta)**. All results are averaged over 20 random seeds. The shade indicates standard error.

comparison, at each environment time step, we stochastically sample the same number of mini-batches to train those model-free baselines as the number of planning updates in Dyna variants. We are able to fix the same HC hyperparameter setting across all environments. Whenever it involves an online learned model, we use the mean squared error to learn a deterministic model, which we found to be reasonably good on those tested domains in this paper. Please see Appendix A.8 for experiment details.[2] We also refer readers to Appendix A.7.4 for experiments on the autonomous driving domain.

**Performances on benchmarks.** Figure 4 shows the performances of different algorithms on MountainCar, Acrobot, GridWorld (Figure 2(a)), and CartPole. On these small domains, we focus on studying our sampling distribution and hence we need to isolate the effect of model errors (by using a true environment model), though we include our algorithm Dyna-TD with an online learned model for curiosity. We have the following observations. First, our algorithm Dyna-

TD consistently outperforms PrioritizedER across domains and planning updates. In contrast, the PrioritizedER may not even outperform regular ER, as occurred in the previous supervised learning experiment.

Second, Dyna-TD's performance significantly improves and even outperforms other Dyna variants when increasing the planning budget (i.e., planning updates $n$) from 10 to 30. This validates the utility of those additional hypothetical experiences acquired by our sampling method. In contrast, both ER and PrioritizedER show limited gain when increasing the planning budget (i.e., number of mini-batch updates), which implies the limited utility of those visited experiences.

**Dyna variants comparison**. Dyna-Value occasionally finds a sub-optimal policy when using a large number of planning updates, while Dyna-TD always finds a better policy. We hypothesize that Dyna-Value results in a heavy sampling distribution bias even during the late learning stage, with density always concentrated around the high-value regions. We verified our hypothesis by checking the entropy of the sampling distribution in the late training stage, as shown in Figure 5. A high entropy indicates the sampling distri-

---

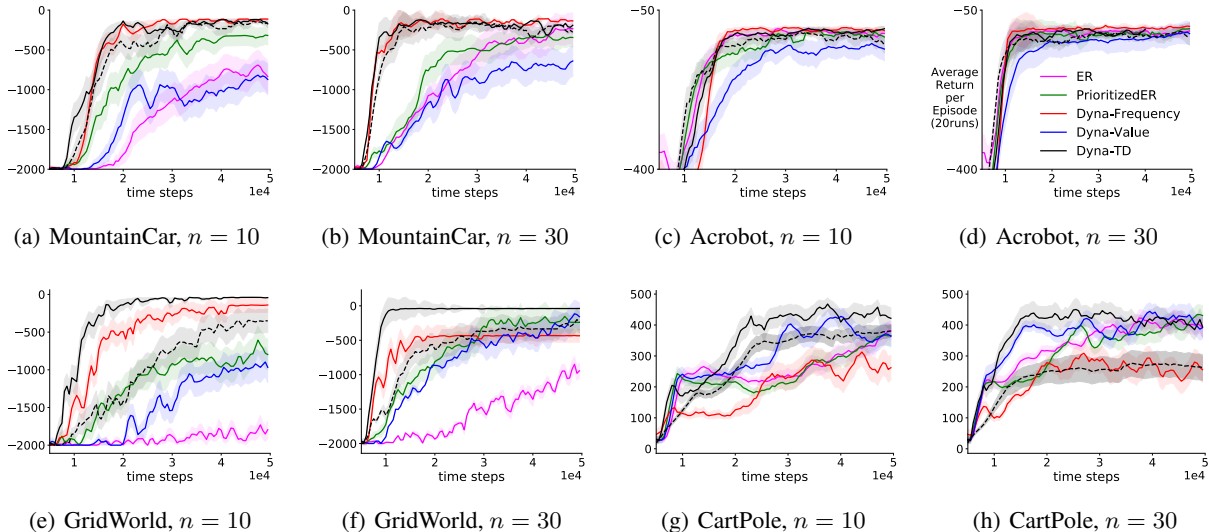

(a) MountainCar, $n = 10$    (b) MountainCar, $n = 30$    (c) Acrobot, $n = 10$    (d) Acrobot, $n = 30$

(e) GridWorld, $n = 10$    (f) GridWorld, $n = 30$    (g) CartPole, $n = 10$    (h) CartPole, $n = 30$

Figure 4: Episodic return v.s. environment time steps: evaluation learning curves of **Dyna-TD (black)**, **Dyna-Frequency (red)**, **Dyna-Value (blue)**, **PrioritizedER (forest green)**, and **ER(magenta)** with planning updates $n = 10, 30$. The **dashed** line denotes Dyna-TD with an online learned model. All results are averaged over 20 random seeds after smoothing over a window of size 30. The shade indicates standard error. Results with planning updates $n = 5$ are in Appendix A.7.3.

bution is more dispersed than the one with low entropy. We found that the sampling distribution of Dyna-Value has lower entropy than Dyna-TD.

Dyna-Frequency suffers from explosive or zero gradients. It requires computing third-order differentiation $\nabla_s ||H_v(s)||$ (i.e., taking the gradient of the Hessian). It is hence sensitive to domains and parameter settings such as learning rate choice and activation type. This observation is consistent with the description from Pan et al. [2020].

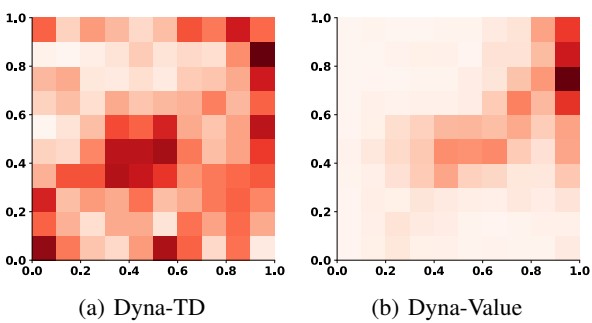

(a) Dyna-TD    (b) Dyna-Value

Figure 5: Sampling distributions on the GridWorld visualized by building 2D histogram from sampled states. Heavy color indicates high visitations/state density. The concrete way of generating the distribution is the same as Figure 2. (a) has entropy around $4.5$ and (b) has entropy around $3.9$.

**A demo for continuous control.** We demonstrate that our approach can be applied for Mujoco [Todorov et al., 2012] continuous control problems with an online learned model and still achieve superior performance. We use DDPG (Deep Deterministic Policy Gradient) [Lillicrap et al., 2016, Silver et al., 2014] as an example for use inside our Dyna-TD. Let

$\pi_{\theta'} : \mathcal{S} \mapsto \mathcal{A}$ be the actor, then we set the HC function as $h(s) \stackrel{\text{def}}{=} \log |\hat{y} - Q_\theta(s, \pi_{\theta'}(s))|$ where $\hat{y}$ is the TD target. Figure 6 (a)(b) shows the learning curves of DDPG trained with ER, PrioritizedER, and our Dyna-TD on Hopper and Walker2d respectively. Since other Dyna variants never show an advantage and are not relevant to the purpose of this experiment, we no longer include them. Dyna-TD shows quick improvement as before. This indicates our sampled hypothetical experiences could be helpful for actor-critic algorithms that are known to be prone to local optimums. Additionally, we note again that ER outperforms PrioritizedER, as occurred in the supervised learning (PrioritizedL2 is worse than L2) experiments.

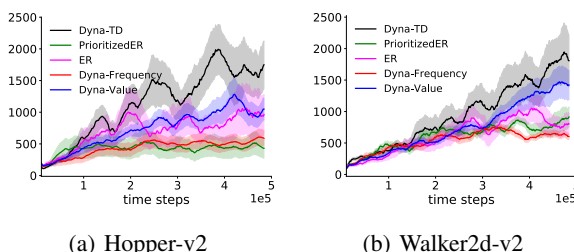

(a) Hopper-v2    (b) Walker2d-v2

Figure 6: Episodic returns v.s. environment time steps of **Dyna-TD (black)** with an online learned model, and other competitors. Results are averaged over 5 random seeds after smoothing over a window of size 30. The shade indicates standard error.

## 6 DISCUSSION

We provide theoretical insight into the error-based prioritized sampling by establishing its equivalence to the uniform

sampling for a cubic power objective in a supervised learning setting. Then we identify two drawbacks of prioritized ER: outdated priorities and insufficient sample space coverage. We mitigate the two limitations by SGLD sampling method with empirical verification. Our empirical results on both discrete and continuous control domains show the efficacy of our method.

There are several promising future directions. First, a natural question is how a model should be learned to benefit a particular sampling method, as this work mostly focuses on sampling hypothetical experiences without considering model learning algorithms. Existing results show that learning a model while considering how to use it should make the policy robust to model errors [Farahmand et al., 2017, Farahmand, 2018]. Second, one may apply our approach with a model in latent space [Hamilton et al., 2014, Wahlström et al., 2015, Ha and Schmidhuber, 2018, Hafner et al., 2019, Schrittwieser et al., 2020], which enables our method to scale to large domains. Third, since there are existing works examining how ER is affected by bootstrap return [Daley and Amato, 2019], by buffer or mini-batch size [Zhang and Sutton, 2017, Liu and Zou, 2017], and by environment steps taken per gradient step [Fu et al., 2019, van Hasselt et al., 2018, Fedus et al., 2020], it is worth studying the theoretical implications of those design choices and their effects on prioritized ER's efficacy.

Last, as our cubic objective explains only one version of the error-based prioritization, efforts should also be made to theoretically interpret other sampling distributions, such as distribution location or reward-based prioritization [Lambert et al., 2020]. It is interesting to explore whether these alternatives can be formulated as surrogate objectives. Furthermore, a recent work by Fujimoto et al. [2020] establishes an equivalence between various distributions and uniform sampling for different loss functions. Studying if those general loss functions have faster convergence rate as shown in our Theorem 2 could help illuminate their benefits.

## Acknowledgements

We would like to thank all anonymous reviewers for their helpful feedback during multiple submissions of this paper. We acknowledge the funding from the Canada CIFAR AI Chairs program, Alberta Machine Intelligence Institute, and Natural Sciences and Engineering Council of Canada (NSERC) Discovery Grant.

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
