# OpenReview forum: "Understanding and Mitigating the Limitations of Prioritized Experience Replay"
_auai.org/UAI/2022/Conference — UAI 2022 Poster_

### Official Review · Reviewer_sNne · 2022-04-12

**Q2(1) Originality/Novelty:** 3
**Q2(2) Significance/Impact:** 3
**Q2(3) Correctness/Technical Quality:** 3
**Q2(6) Clarity Of Writing:** 3
**Q6 Overall Score:** 7
**Q8 Confidence In Your Score:** 3

**Q1 Summary And Contributions:**

This paper has two main contributions: (i) it shows theoretically and experimentally two fundamental issues with Prioritized Experience Replay ("outdated priorities" and "insufficient coverage of sample space"); (ii) it proposes a novel model-based sampling approach for overcoming those issues. The approach builds upon the HC-Dyna algorithm. Several investigations support their arguments, including synthetic experiments, simulation experiments (including Open AI Gym domains), and theorems.

**Q2 Assessment Of The Paper:**

More detailed information regarding each of these aspects is given below:

**Q2(4) Quality Of Experiments (Optional):**

4: Excellent: The experimental evaluation is comprehensive and the results are compelling.

**Q2(5) Reproducibility:**

2: Fair: Key resources (e.g., proofs, code, data) are unavailable but key details (e.g., proof sketches, experimental setup) are sufficiently well-described for an expert to confidently reproduce the main results.

**Q3 Main Strengths:**

- The paper builds upon an existing framework (Dyna), but the sampling mechanism and the way it is applied to improve DQN and DDPG seems novel. Additionally, the detailed analysis of issues with Prioritised Experience replay also seems novel.

- The experience replay buffer is quite an importance piece of DQN and several other derived algorithms. Hence, I believe this paper would have a good impact in the AI community.

- The work seems technically correct, as far as I can tell. It is also nice how the argument is based on synthetic experiments, mathematical proofs, and experiments in simulation.

- The experiments are quite comprehensive, and support well the main claims. It is nice how the fundamental issues with ER are first demonstrated in a simple supervised learning setting, and then further validated in simulation experiments.

- The appendix includes detailed proofs, and explanations on how to reproduce the experiments. Pseudo-code is available.

-  The paper is well-written.

**Q4 Main Weakness:**

Originality/Novelty: N/A

Significance/Impact: N/A

Correctness/Technical Quality: I have a few questions under Q5 below.

Quality Of Experiments (Optional): It is nice that the experiments show the standard error. However, the work would be stronger with tests of statistical significance.

Reproducibility: Source code does not seem to be available, so it may take quite some effort to reproduce the approach, although pseudo-code and extensive details are available in the appendix.

Clarity Of Writing: I include some suggestions in Q5.

**Q5 Detailed Comments To The Authors:**

- Page 2, MBRL and Dyna: "We consider a one-step model which maps a state-action pair to its possible next state and reward: P : S × A → S × R." -> I got confused here, since we are handling MDPs shouldn't the map be to a distribution across states and rewards? The model assumes a deterministic transition function?

- Page 5, Large sample space coverage: "These visualizations verify that our sampled states cover better the sample space than the prioritized ER does." -> However, is it really worth it exploring these states? If they are far from the shortest path towards the target, then perhaps it is better to just prune them out and not explore them much, in order to focus the computation on more valuable states?

- Section 4.1, and equation 5. I am somewhat confused by the update rule. Are we assuming here some vectorial representation of the state, and are numerically updating the feature values directly? If so, how can we know that the resulting states are valid?

- Suggestions for the writing/presentation:

    - "We then empirically study the such limitations." -> "We then empirically study such limitations." or "We then empirically study the limitations."

    - "Setting ϵ = 570(i.e., ϵ(i) ≈ 0.57)" -> Missing space after 570.

    - "In fact, in RL, “all training samples” in RL are restricted to those visited experiences in the ER buffer" -> "in RL" repeated.

    - "Previous work Pan et al. [2020] show" -> Wrong reference format.

    - Figure 1 (and other similar ones): It would be better to have something to aid visualisation in addition to the colors. When reading in gray-scale, it is not possible to differentiate the lines.

    - "Third, since there are existing works examining how ER is affected by bootstrap return [Daley and Amato, 2019], by buffer or mini-batch size [Zhang and Sutton, 2017, Liu and Zou, 2017], and by environment steps taken per gradient step [Fu et al., 2019, van Hasselt et al., 2018, Fedus et al., 2020]. It is worth studying the theoretical implications of those design choices and their effects on prioritized ER’s efficacy." -> "Third, since there are existing works examining how ER is affected by bootstrap return [Daley and Amato, 2019], by buffer or mini-batch size [Zhang and Sutton, 2017, Liu and Zou, 2017], and by environment steps taken per gradient step [Fu et al., 2019, van Hasselt et al., 2018, Fedus et al., 2020], it is worth studying the theoretical implications of those design choices and their effects on prioritized ER’s efficacy."

    - "Section A.5: the theorem characterizing the error bound between the sampling distribution estimated by using a true model and a learned model. It includes the the full proof." -> the the

= Rebuttal =

I read your reply, thanks for the clarifications.

**Q7 Justification For Your Score:**

I found this work quite interesting and complete. I think detailed studies on the Experience Replay can have quite a nice impact, and they contribute further with a new sampling technique.

I do have a few questions about the methodology, which hopefully can be clarified.

**Q9 Complying With Reviewing Instructions:**

1: Yes.

---

### Official Review · Reviewer_qKpW · 2022-04-13

**Q2(1) Originality/Novelty:** 3
**Q2(2) Significance/Impact:** 2
**Q2(3) Correctness/Technical Quality:** 3
**Q2(6) Clarity Of Writing:** 2
**Q6 Overall Score:** 5
**Q8 Confidence In Your Score:** 3

**Q1 Summary And Contributions:**

The paper provides theoretical insights into error-based prioritized sampling by showing its equivalence to uniform sampling for a cubic power loss function. After that, the paper discusses 2 limitations: outdated priorities and sample space coverage. The paper then tackles the limitations by proposing the SGLD sampling method. The experiments show the effectiveness of the proposed sampling technique.

**Q2 Assessment Of The Paper:**

More detailed information regarding each of these aspects is given below:

**Q2(4) Quality Of Experiments (Optional):**

3: Good: The experimental evaluation is adequate, and the results convincingly support the main claims.

**Q2(5) Reproducibility:**

3: Good: Key resources (e.g., proofs, code, data) are available and key details (e.g., proofs, experimental setup) are sufficiently well-described for competent researchers to confidently reproduce the main results.

**Q3 Main Strengths:**

The paper discusses the related work appropriately, and discusses the shortcoming of prioritized experience replay sampling methods using the theoretical analysis. The paper then tackles the issues effectively and demonstrates its effectiveness on few applications.

**Q4 Main Weakness:**

1) The paper is hard to read and would likely benefit a small number of researchers working on this specific topic.
2) There is an issue with the pdf file. It keeps crashing and is slow to scroll. I tried it on 2 different machines, and the issue still persists.



**Q5 Detailed Comments To The Authors:**

1) In Figure 1 a, the y axis description appears to be missing.
2) Have you tried other baselines: PPO, TRPO, DDPG, how do they compare to the proposed method?

**Q7 Justification For Your Score:**

The paper addresses a detail in RL and shows some interesting findings. The effectiveness of the approach is shown on toy problems (e.g. Mountain Car, Hopper, Walker2d). Some of the important baselines appear to be missing: PPO, TRPO, DDPG.

**Q9 Complying With Reviewing Instructions:**

1: Yes.

---

### Official Review · Reviewer_CrF8 · 2022-04-13

**Q2(1) Originality/Novelty:** 2
**Q2(2) Significance/Impact:** 3
**Q2(3) Correctness/Technical Quality:** 3
**Q2(6) Clarity Of Writing:** 3
**Q6 Overall Score:** 6
**Q8 Confidence In Your Score:** 3

**Q1 Summary And Contributions:**

This paper studies the prioritized experience replay from a theoretical perspective. This paper shows the equivalence between the error-based prioritized sampling method and optimizing a cubic power objective. From this connection, it finds two limitations of prioritized ER methods: outdated priorities and insufficient coverage of sample space. To address the limitations, the authors propose the algorithm Dyna-TD and conduct experiments to show the efficiency of Dyna-TD.

**Q2 Assessment Of The Paper:**

More detailed information regarding each of these aspects is given below:

**Q2(4) Quality Of Experiments (Optional):**

3: Good: The experimental evaluation is adequate, and the results convincingly support the main claims.

**Q2(5) Reproducibility:**

3: Good: Key resources (e.g., proofs, code, data) are available and key details (e.g., proofs, experimental setup) are sufficiently well-described for competent researchers to confidently reproduce the main results.

**Q3 Main Strengths:**

PER is a common and practical trick for off-policy RL methods that have been widely used in many open-source RL libraries. However, it is somewhat a heuristic method and it is necessary to analyze why it works theoretically. This paper starts with the limitations of PER from a theoretical perspective and proposes to address the limitations, which could be helpful to improve the efficiency of the current RL methods.

**Q4 Main Weakness:**

First, the novelty of the proposed method Dyna-TD is limited. Dyna-TD is an instance of the existing algorithmic framework HC-Dyna. Dyna-TD chooses a different HC function and this modification is incremental.

Second, I wonder whether the proposed technique could be incorporated into the current value-based RL methods. If true, the proposed technique could be more general and significant.




**Q5 Detailed Comments To The Authors:**

I think the proposed method belongs to the subfield of ‘how to use ER’. There are some missing related works [1, 2, 3, 4] towards modifying the sampling distribution of ER.

References:

[1] Aviral Kumar, Abhishek Gupta, Sergey Levine Discor: Corrective feedback in reinforcement learning via distribution correction. In NeurIPS 2020.

[2] Samarth Sinha, Jiaming Song, Animesh Garg, and Stefano Ermon. Experience replay with
likelihood-free importance weights. CoRR, abs/2006.13169, 2020.

[3] Kimin Lee, Michael Laskin, Aravind Srinivas, and Pieter Abbeel. SUNRISE: A simple unified framework for ensemble learning in deep reinforcement learning. In ICML 2021.

[4] Xu-Hui, Liu, Zhenghai Xue, Jingcheng Pang, Shengyi Jiang, Feng Xu, and Yang Yu. Regret Minimization Experience Replay in Off-Policy Reinforcement Learning. In NeurIPS 2021.

**Q7 Justification For Your Score:**

The usage of ER is an important direction to improve the effectiveness of off-policy RL methods. This paper focuses on the most popular error-based method and proposes an efficient technique to improve it, which is a contribution to the RL community. Thus, I recommend a weak accept.

**Q9 Complying With Reviewing Instructions:**

1: Yes.

---

### Decision · Program_Chairs · 2022-05-15

**Decision:**

Accept (Poster)

**Comment:**

Meta Review: The reviewers felt that the paper is borderline but just above the acceptance threshold. The key issue that the authors need to address as (1) make the paper more readable (and fix issues with the PDF); (2) Cite relevant related work and (3) address several issues raised by reviewer sNne.

These issues are not hard to address and the authors have addressed them more or less in their response.

In summary, a good paper that can be accepted as a poster (because it is relevant to a smaller sub-community).